# Improving Open-World Classification with Disentangled Foreground and Background Features

## ABSTRACT

Detecting out-of-distribution (OOD) inputs is a principal task for ensuring the safety of deploying deep-neural-network classifiers in open-world scenarios. OOD samples can be drawn from arbitrary distributions and exhibit deviations from in-distribution (ID) data in various dimensions, such as foreground features (e.g., objects in CIFAR100 images vs. that in CIFAR10 images) and background features (e.g., textural images vs. objects in CIFAR10). Existing methods can confound foreground and background features in training, failing to utilize the background features for OOD detection. This paper considers the importance of feature disentanglement in open-world classification and proposes the simultaneous exploitation of both foreground and background features to support the detection of OOD inputs in open-world classification. To this end, we propose a novel framework that first disentangles foreground and background features from ID training samples via a dense prediction approach, and then learns a new classifier that can evaluate the OOD scores of test images from both foreground and background features. It is a generic framework that allows for a seamless combination with various existing OOD detection methods. Extensive experiments show that our approach 1) can substantially enhance the performance of four different state-of-the-art (SotA) OOD detection methods on multiple widely-used OOD datasets with diverse background features, and 2) achieves new SotA performance on these benchmarks.

## CCS CONCEPTS

• **Computing methodologies** → *Object recognition.*

## KEYWORDS

Open-World Classification, Out-of-Distribution, Disentangled Feature

## 1 INTRODUCTION

Deep neural networks have demonstrated superior performance in computer vision tasks [24]. Most deep learning methods assume that the training and test data are drawn from the same distribution. Thus, they fail to handle real-world scenarios with out-of-distribution (OOD) inputs that are not present in the training data [35]. Failures in distinguishing these OOD inputs from in-distribution (ID) data may lead to potentially catastrophic decisions,

**Unpublished working draft. Not for distribution.**

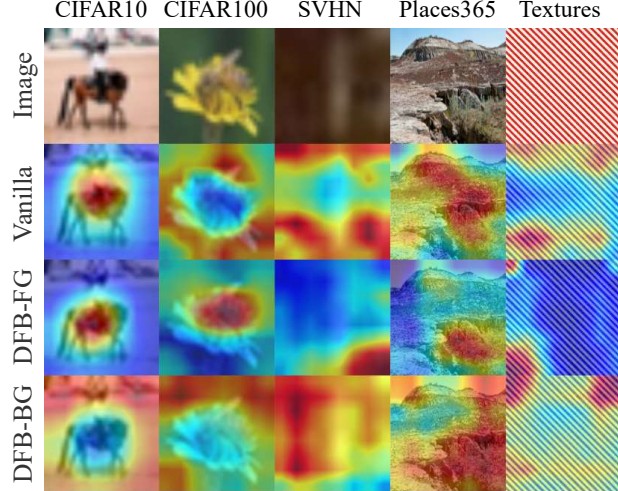

Figure 1: Example images in ID (CIFAR10 [23]) and OOD datasets (CIFAR100 [23], SVHN [32], Places365 [53], Textures [7]) with their attention maps from a vanilla classifier and our proposed DFB classifier. Vanilla classifiers tend to focus on objects unrelated to the ID class, e.g., the person on a horse (ID class), due to spurious correlation. In OOD data, vanilla classifiers struggle to localize objects within the image and treat the background features as foreground for ID classification. By disentangling foreground and background features, DFB effectively addresses these issue.

especially in safety-critical applications like autonomous driving or medical systems [5]. In open-world applications, encountering unexpected OOD inputs is a common occurrence. Consequently, detecting and rejecting these OOD inputs has emerged as a significant challenge in the secure deployment of deep neural networks.

OOD detection approaches are designed to address this problem, which aim to detect and reject these OOD samples while guaranteeing the classification of in-distribution data [16]. There are generally two groups of OOD detection approaches. One of them are post-hoc approaches that work with a trained classification network to derive OOD scores without re-training or fine-tuning of the network, *e.g.*, by using maximum softmax probability of the network outputs [16], maximum logits [15], or the Mahalanobis distance between the input and the class centroids of ID data [26]. Another group of approaches fine-tunes the classifiers with different methods, such as the use of pseudo OOD samples [4, 17, 28, 31, 34, 45, 47]. Most of these methods, especially the post-hoc methods, are primarily based on the "foreground features" to detect OOD samples. These are the features that exhibit the semantics of the in-distribution classes, such as the appearance features of the 'horse' class images in the

CIFAR10 image classification, as shown in Fig. 1. This introduces two problems: 1) Owing to dataset bias, the foreground features learned by vanilla classifiers may develop spurious correlations [30] with other objects in the image. For instance, a correlation may be incorrectly drawn between horses and riders. These spurious correlations introduce irrelevant semantics, which can lead to the misclassification of not only ID images but also OOD images. 2) Focusing on foreground features overlooks other dimensions that could also be important for OOD detection, as OOD samples can be drawn from arbitrary distributions and can exhibit deviations from in-distribution (ID) data in various dimensions. One such dimension is the set of "background features" that exhibit no class semantics. We observe that the regions of interest for the classifiers on OOD samples often tend to be the background, as shown in the CIFAR100 example in Fig. 1. This phenomenon indicates that existing classifiers have a foreground-background confusion issue, and they misclassify OOD samples to the ID classes with similar semantic features to these background features.

This paper considers the importance of disentangling foreground and background features in OOD detection and proposes to leverage background features to enhance the OOD detection methods that are based on foreground features. To this end, we introduce a novel generic framework, called DFB, that can **Disentangle the Foreground and Background features** from ID training samples by a dense prediction approach, with which different existing foreground-based OOD detection methods can be seamlessly combined to learn the in-distribution features from both the foreground and background dimensions. Specifically, given a trained $K$-class classification network where $K$ is the number of in-distribution classes, DFB first generates pseudo semantic segmentation masks by a weakly-supervised segmentation approach that uses image-level labels to locate discriminative regions in the images. These pseudo segmentation masks are then utilized to train a $(K+1)$-class dense prediction network, *with the first $K$ classes being the original $K$ ID classes and the $(K+1)$-th class corresponding to the ID background features*. The dense prediction network is further converted into a $(K+1)$-class classification network by adding a global pooling layer. The conversion is lossless and requires no re-training. In doing so, the $(K+1)$-class classifier learns both foreground and background ID features. Different existing foreground-based methods, such as the post-hoc methods, can be applied to the first $K$ prediction outputs to obtain **semantic OOD scores**, while the $(K+1)$-th prediction can be directly used to define **background OOD scores**. Combining these semantic and background OOD scores enable OOD detection from both foreground and background features.

As depicted in Fig. 1, the proposed DFB effectively disentangles foreground and background features. In the ID data CIFAR10, DFB can more accurately locate the ID objects than the vanilla classifier. In the OOD data CIFAR100, which contains significant foreground objects, DFB successfully disentangle between the foreground and background objects, both of which are important for detecting the OOD. In the other three OOD datasets without prominent foreground objects, the foreground branch of DFB focuses on fewer areas compared to the vanilla classifier while its background branch recognizes most of the background areas, in which the background OOD scores would exert greater influence on the OOD detection. Note that the influence of semantic and background features is

determined by the area of these features, so a hyperparameter with a fixed value can well control these two OOD scores across diverse OOD datasets; no careful tuning of this hyperparameter is required per OOD dataset.

In summary, we make the following main contributions:

- This work studies the importance of disentangling foreground and background features and proposes to synthesize both foreground and background features for more effective OOD detection in diverse real-world applications. This provides a new insight into the OOD detection problem.
- We then propose a novel approach DFB, in which different existing foreground-based OOD detection methods can be seamlessly combined to jointly learn the ID features from both foreground and background dimensions. It offers a generic approach to enhance current OOD detection methods. To our knowledge, this is the first generic framework for joint foreground and background OOD detection.
- Extensive experiments on four widely-used OOD datasets with diverse background show that our approach DFB 1) can substantially enhance the performance of four different state-of-the-art (SotA) OOD detection methods, and 2) achieves new SotA performance on these benchmarks.

## 2 RELATED WORK

**Post-hoc Approaches.** Modelling the uncertainty of pre-trained DNN directly without retraining the network is one popular approach for OOD detection [2, 8, 10, 12, 20, 29, 33, 38, 43, 48, 54]. Hendrycks et al. [16] propose the uncertainty of DNNs and establish a baseline for OOD detection by maximum softmax probability (MSP). ODIN [27] introduces input perturbation and temperature scaling to enhance MSP. Lee et al. [26] propose the deep Mahalanobis distance-based detectors, which compute the distance-based OOD scores from the pre-trained networks' features. Liu et al. [28] calculate the logsunexp on logit as the energy OOD score. ReAct [39] reduces the DNN's overconfidence in OOD samples by activation clipping, which further enhances the energy scores. MaSF [13] considers the empirical distribution of each layer and channel in the CNN and returns a p-value as the OOD score. ViM [42] attempts to utilize not only primitive semantic features but also their residuals to define more effective logit-based OOD scores. Recently, DML [51] reformulate the logit into cosine similarity and logit norm and propose to use flexibility balanced MaxCosine and MaxNorm. This type of approach relies on the foreground semantic features learned by the pre-trained networks, which neglects other relevant features, such as background features.

**Fine-tuning Approaches.** Another dominant approach is to fine-tune the classification networks for adapting to the OOD detection tasks. In this line of research, Hsu et al. [19] further improve ODIN by decomposing confidence scoring. Zaeemzadeh et al. [49] project ID samples into a one-dimensional subspace during training. Some other studies [21, 41] group ID data and assume them as OOD samples for each other to guide the network training. Some studies [3, 37] have noticed the influence of background features on OOD detection, but they focus on solving the confusion between background and semantic features, and ignore the positive influence of background features on OOD detection itself. Outlier Exposure

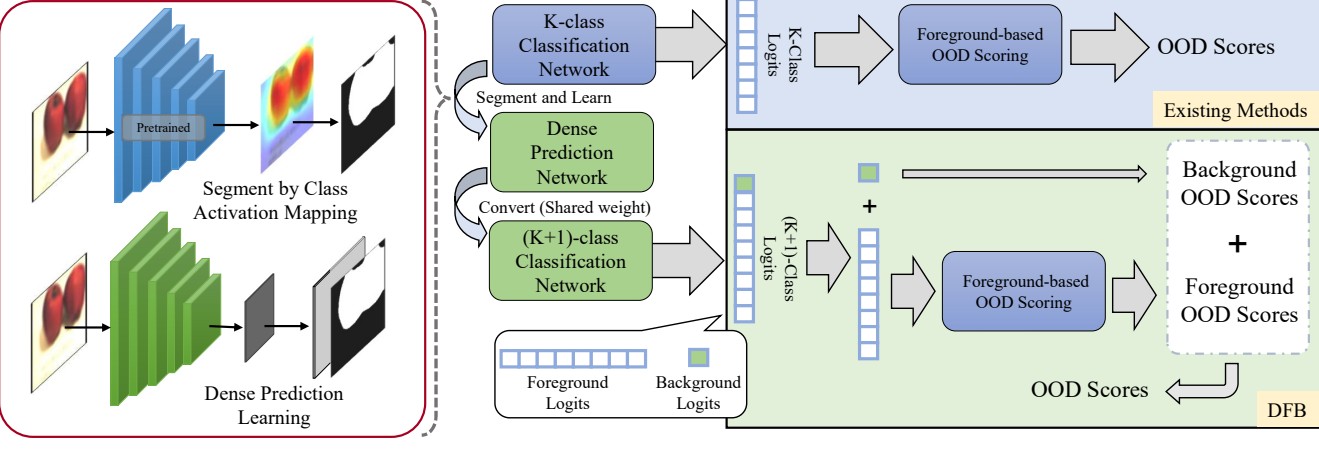

**Figure 2: Overview of our proposed framework. It first uses a trained $K$-class classification network to obtain pseudo semantic segmentation masks and then learns the in-distribution features by training a $(K+1)$-class classification network with the pseudo labels (Left). It lastly converts the dense prediction network to a $(K+1)$-class classifier in a lossless fashion, and leverages these $(K+1)$ prediction outputs for joint foreground and background OOD detection (Bottom Right).**

(OE) [17] introduces auxiliary outlier data to train the network and improve its OOD detection performance. Such approaches can use real outliers [4, 6, 28, 31, 34, 45, 47] or synthetic ones from generative models [25]. The performance of this approach often depends on the quality of the outlier data. Fine-tuning the networks may also lead to the loss of semantic information and consequently degraded ID classification accuracy.

## 3 PROPOSED APPROACH

**Problem Statement.** Given a set of training samples $\mathcal{X} = \{\mathbf{x}_i, \mathbf{y}_i\}_{i=1}^{N}$ drawn from an in-distribution $P_{\mathcal{X}}$ with label space $\mathcal{Y} = \{y_j\}_{j=1}^{K}$, and let $f : \mathcal{X} \to \mathbb{R}^{\mathcal{Y}}$ be a classifier trained on the in-distribution samples $\mathcal{X}$, then the goal of OOD detection is to obtain a new decision function $g$ to discriminate whether $\mathbf{x}$ come from $P_{\mathcal{X}}$ or out-of-distribution data $P_{out}$:

$$g(\mathbf{x}, f) = \begin{cases} 1 & \text{if } \mathbf{x} \in P_{out}, \\ 0 & \text{if } \mathbf{x} \in P_{\mathcal{X}}. \end{cases}$$

The difference between $P_{\mathcal{X}}$ and $P_{out}$ determines the difficulty of detecting the OOD samples. Existing OOD detection approaches focus on the difference between $P_{\mathcal{X}}$ and $P_{out}$ based on the semantic information of the class label space $\mathcal{Y}$, neglecting other relevant dimensions such as the background feature space. This work aim to learn the background features and leverage them to complement these foreground-based OOD detection approaches.

## 3.1 Overview of Our Approach

Using semantic of foreground objects only to detect OOD samples can often be successful when the OOD samples have some dominant semantics that are different from the ID images. However, this type of approach would fail to work effectively when the OOD samples do not have clear object semantics and/or exhibit some similar semantic appearance to the ID samples, *e.g.*, the images illustrated in Fig. 1. Motivated by this, we introduce a generic framework DFB, in which the model disentangle the foreground and background features of the in-distribution data and learns ID background features, upon which different existing OOD detection methods can be applied with the learned background representations to detect OOD samples from both of the foreground and background dimensions. A high-level overview of our proposed framework is provided in Fig. 2.

(1) DFB first disentangles and learns the in-distribution foreground and background features by a $(K+1)$-class dense prediction network trained from the given pre-trained K-class classification network.

(2) It then seamlessly integrates the foreground and background features into image classification models by transforming the dense prediction network to a $(K+1)$-class classification network, where the prediction entries of the $K$ classes are focused on the class semantics of the $K$ in-distribution class while the extra (+1) class is focused on the in-distribution background features.

(3) Lastly, an OOD score in the foreground dimension obtained from existing post-hoc OOD detectors based on the $K$-class predictions, and an OOD score obtained from the extra (+1) class prediction from the background dimension, are synthesized to perform OOD detection.

## 3.2 Learning In-distribution Background Features via $(K+1)$-class Dense Prediction

DFB aims to learn distinct representations of foreground and background information in images, while also considering them as in-distribution features. The key challenge here is how to locate these

background features and separate them from the foreground features. We introduce a weakly-supervised dense prediction method to tackle this challenge, in which weakly-supervised semantic segmentation methods are first utilized to generate pseudo segmentation mask labels that are then used to train a $(K + 1)$-class dense prediction network. The extra (+1) class learned in the dense predictor is specifically designed to learn the background features, while the other $K$ class predictions are focused on learning the foreground features of the $K$ classes given in the training data. Particularly, given the training data $\mathcal{X}$ with $K$-class image-level labels $\mathcal{Y}$ and a trained $K$-class classification network $\phi$, the pseudo segmentation mask labels can be obtained by the class activation mapping [52]:

$$\mathbf{M}_{y_{\mathbf{x}}}^{(i,j)} = \mathbf{W}_{y_{\mathbf{x}}}^{\top} \phi_{\text{cnn}}(\mathbf{x})^{(i,j)}, \tag{1}$$

where $\mathbf{W}_{y_{\mathbf{x}}}$ is the classification weight of the trained classifier $\phi$ corresponding to the groundtruth class $y_{\mathbf{x}}$ of $\mathbf{x}$, and $\phi_{\text{cnn}}(\mathbf{x})^{(i,j)}$ obtains the feature vector at the unit $(i, j)$ in the feature map extracted by the feature extractor in $\phi$ from image $\mathbf{x}$. $\mathbf{M}_{y_{\mathbf{x}}} \in \mathbb{R}^{H \times W}$ is an attention map indicating a pixel-wise semantic score of $\mathbf{x}$ relative to its groundtruth class $y_{\mathbf{x}}$. We then define a foreground decision threshold $\theta$ to generate the fine-grained pseudo labels of background pixels and foreground pixels by:

$$\hat{\mathbf{Y}}^{(i,j)}(\mathbf{x}) = \begin{cases} 0 & \text{if } \mathbf{M}_{y_{\mathbf{x}}}^{(i,j)} < \theta, \\ 1 & \text{if } \mathbf{M}_{y_{\mathbf{x}}}^{(i,j)} \geqslant \theta, \end{cases} \tag{2}$$

where the attention scores are normalized into the range $[0, 1]$ and $\theta = 0.5$ is used. We then leverage these pseudo labels of the foreground and background pixels, $\hat{\mathbf{Y}}$, to train a $(K + 1)$-class dense prediction network $f_{\Theta_d} : \mathcal{X} \to \{0, 1\}^{K \times H \times W}$ via a pixel-level cross entropy loss:

$$L(\mathbf{x}, \hat{\mathbf{Y}}) = \frac{-1}{H \times W} \sum_{i=1}^{H} \sum_{j=1}^{W} \sum_{k=1}^{K+1} \hat{y}_k^{(i,j)} \log \left( f(\mathbf{x}, \Theta_d)_k^{(i,j)} \right), \tag{3}$$

where $f(\mathbf{x}, \Theta_d)^{(i,j)}$ outputs a prediction vector consisting of prediction probabilities of the $K + 1$ classes at the image pixel $(i, j)$, and $\hat{\mathbf{y}}^{(i,j)}$ denotes the corresponding pseudo labels at the same pixel. In doing so, $f_{\Theta_d}$ learns both in-distribution foreground and background features.

## 3.3 Dense Prediction to Image Classification

The pixel-level foreground and background features learned in the dense prediction network cannot be applied directly to the image classification task. We show below that the $(K + 1)$-class dense prediction network can be transformed to a $(K + 1)$-class image classification network in a lossless fashion: the dense prediction and the classification networks share the same weight parameters, and the classification network can be applied to image classification without re-training. Particularly, the dense prediction network $f_{\Theta_d} : \mathcal{X} \to \{0, 1\}^{(K+1) \times H \times W}$ can be decomposed into three main modules: 1) a feature extraction network $f_{\Theta_{CNN}} : \mathcal{X} \to \mathcal{G}$ consisting of a convolutional neural network that extracts the input image $\mathbf{x} \in \mathbb{R}^{3 \times H \times W}$ into a smaller scale but larger dimensional feature map $\mathbf{G} \in \mathbb{R}^{C \times h \times w}$, 2) an upsampling module up$(\cdot)$ that upsamples the feature map $\mathbf{G}$ to original input size $H \times W$, typically implemented using bilinear interpolation, and 3) a 1x1 convolution

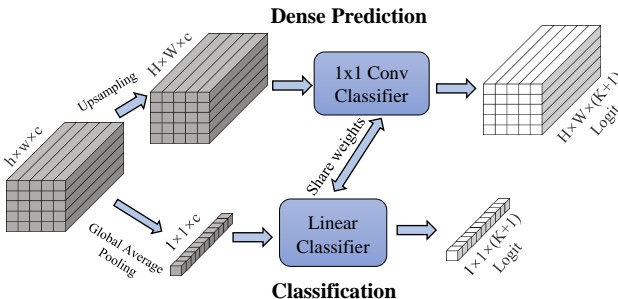

**Figure 3: Lossless conversion of a dense prediction network to a classification network.**

classifier $f_{\Theta_{cls}} : \mathcal{G} \to \mathcal{L}$ that computes the logit for each pixel in the feature map and outputs a logit map $\mathbf{L} \in \mathbb{R}^{(K+1) \times H \times W}$. The size of weights of the convolutional classifier is $C \times (K + 1)$. Thus, the dense prediction network $f_{\Theta_d}$ can be denoted as:

$$f(\mathbf{x}, \Theta_d) = \text{softmax}(f(\text{up}(f(\mathbf{x}, \Theta_{CNN})), \Theta_{cls})), \tag{4}$$

where $\Theta_d = \{\Theta_{CNN}, \Theta_{cls}\}$.

For a classification network $f_{\Theta_c} : \mathcal{X} \to \mathbb{R}^{\mathcal{Y}}$, it can be similarly decomposed into three main modules: 1) a feature extraction network $f_{\Theta_{CNN}} : \mathcal{X} \to \mathcal{G}$ with the same function as the dense prediction network, 2) a global average pooling GAP$(\cdot)$, which compresses the feature map of $C \times H \times W$ into a feature vector of size $C \times 1 \times 1$, integrating the features of the full image, and 3) a linear classifier $f_{\Theta_{cls}} : \mathcal{G} \to \mathcal{L}$, which computes the logit of the full image based on the feature vector with $C \times (K + 1)$ weights. The classification network $f_{\Theta_c}$ can be denoted as:

$$f(\mathbf{x}, \Theta_c) = \text{softmax}(f(\text{GAP}(f(\mathbf{x}, \Theta_{CNN})), \Theta_{cls})), \tag{5}$$

where $\Theta_c = \{\Theta_{CNN}, \Theta_{cls}\}$. It is clear from Eqs. (4) and (5) that the only difference between $f_{\Theta_d}$ and $f_{\Theta_c}$ is the upsampling module and the GAP module, sharing the same feature extraction network and the classifier. Further, the upsampling and the GAP modules are weight-free operations, which can be easily replaced with each other, as shown in Fig. 3. In this way, we directly transfer the in-distribution foreground and background features learned in the dense prediction network $f_{\Theta_d}$ to the image classification network $f_{\Theta_c}$ without any loss of the parameters learned in $f_{\Theta_d}$. For a given test image, the classifier $f_{\Theta_c}$ yields a $(K + 1)$-dimensional logit vector, where the $(K + 1)$-th logit is focused on the in-distribution background features and can be used directly to detect OOD samples from the background dimension:

$$S_b(\mathbf{x}) = \mathbf{L}_{\mathbf{x}}[K + 1], \tag{6}$$

where $\mathbf{L}_{\mathbf{x}} = f(\text{GAP}(f(\mathbf{x}, \Theta_{CNN})), \Theta_{cls})$ is a (K+1)-dimensional prediction logit vector yielded by $f_{\Theta_c}$.

## 3.4 Joint Foreground and Background OOD Detection

Although the background-based OOD score $S_b$ can be used to detect OOD samples directly, it can miss the OOD samples whose detection

relies heavily on the foreground features. Thus, we propose to utilize this background OOD score to complement existing SotA foreground-based OOD detectors. Particularly, since the first $K$ classification logits in $f_{\Theta_c}$ capture similar class semantics as the original $K$-class classifier, off-the-shelf *post-hoc* OOD detection methods that derive an OOD score from these $K$ classification logits can be plugged into DFB to obtain an OOD score from the foreground feature aspect. These foreground and background-based OOD scores are synthesized to achieve a joint foreground and background OOD detection.

There are generally two types of post-hoc OOD detection approaches, including raw logit-based and softmax probability-based methods. Our background-based OOD score is based on an unbounded logit value, which can dominant the overall OOD score when combining with the foreground-based OOD score using the softmax output (its value is within $[0, 1]$). To avoid this situation, we take a different approach to combine the foreground and background-based OOD scores, depending on the type of the foreground-based OOD detector used:

$$S(\mathbf{x}) = \begin{cases} S_h(\mathbf{x}) + \frac{\log(S_b(\mathbf{x}))}{T} & \text{if } S_h \text{ is softmax-based,} \\ S_h(\mathbf{x}) + \frac{S_b(\mathbf{x})}{T} & \text{if } S_h \text{ is logit-based,} \end{cases} \quad (7)$$

where $S(\mathbf{x})$ is the final OOD score used to perform OOD detection in DFB, $S_h(\mathbf{x}) = h(\mathbf{x})$ denotes the OOD score obtained from using an existing foreground-based OOD scoring function $h$, and $T$ is a temperature coefficient hyperparameter. In Eq. (7), to obtain faithful foreground-and-background-combined OOD score, the log function is used to constrain the value and the variance of the background scores $S_b$, while $T$ is used to adjust the distribution of the background scores to match that of the foreground scores.

## 4 EXPERIMENTS

**Datasets.** Following [13, 16, 26–28, 51], we choose two widely used classification datasets: CIFAR10 and CIFAR100 [23], as the in-distribution datasets. As OOD samples are unknown during training, their respective training and test data are used as ID data, with samples from a different dataset added into the test set as the OOD data. To evaluate the effectiveness of our approach, four commonly-used OOD datasets consisting of natural image datasets with diverse background features are used, including SVHN [32], Places365 [53], Textures [7], and CIFAR100/CIFAR10 [23] (CIFAR100 is used as OOD data when CIFAR10 is used as ID data, and vise versa [11, 36, 38]) SVHN is a digit classification dataset cropped from pictures of house number plates, Places365 is a large-scale scene classification dataset,while Textures contains 5,640 texture images in the wild that do not contain specific objects and backgrounds. Images in all these three datasets exhibit largely different foreground and background distributions, so the three datasets contain strong out-of-distribution semantic and background features. On the other hand, both CIFAR10 and CIFAR100 are sampled from Tiny Images [40]and they share similar background features, so when they are used as OOD data for each other, the background features are weak. As a result, this pair of mutual OOD/ID combination is considered as hard OOD detection benchmarks [44]

**Implementation Details.** We use BiT-M [22], a variant of ResNetv2 architecture [14], as the default network backbone throughout the experiments. The official release checkpoint of BiT-M-R50x1 trained on ImageNet-21K is used as our initial $K$-class in-distribution classification model. The model is further fine-tuned on the in-distribution dataset (CIFAR10/CIFAR100) with 20,000 steps using a batch size of 128. SGD is used as the optimizer with an initial learning rate of 0.003 and a momentum of 0.9. We decay the learning rate by a factor of 10 at 30%, 60%, and 90% of the training steps. All images were resized to 160x160 and randomly cropped to 128x128. The Mixup [50] with $\alpha = 0.1$ is also used to synthesize new image samples during training. We subsequently use CAM (Class Activation Mapping) [52] to generate the pseudo mask labels for each in-distribution image based on a multi-scale masking method used in [1] (see Appendix A.1 for the example of pseudo mask and Appendix C.4 for analysis of mask quality). With these pseudo mask labels, we then use a modified Dense-BiT architecture to train the $(K + 1)$-class dense prediction model with the BiT-M-R50x1 checkpoints as the initial weights. All input images are resized to 128x128 during training and inference. We replace the Mixup augmentation used in the training with randomly scaling (from 0.5 to 2.0) and randomly horizontally flipping augmentation. The other training strategy and hyperparameters are maintained the same as the ones used in training the $K$-class classification network above. After that, the dense prediction model is converted to $(K + 1)$-class image classification model using Eq. (5).

Four *post-hoc* OOD detection methods, including MSP [16], ODIN [27], Energy [28], and ViM [42], are used as the plug-in base models. They are respectively employed to combine with DFB to detect OOD samples in both of foreground and background features. To have fair and straightforward comparison, these four plug-in models are built upon the same $K$-class classification model as DFB. The temperature $T = 2.5$ is used in Eq. (7) by default.

We will release our code upon paper acceptance.

**Evaluation Metrics.** We use three widely-used evaluation metrics for OOD detection, including: 1) **FPR95** that evaluates the false positive rate of the OOD samples when the true positive rate of the in-distribution samples is 95%, 2) **AUROC** denotes the Area Under the Receiver Operating Characteristic curve, and 3) **AUPR** is the Area under the Precision-Recall curve. The ID images are the positive samples in calculating AUROC and AUPR to measure the OOD detection performance. In addition, we also report the **Top-1 accuracy** of classifying the in-distribution samples.

### 4.1 Main Results

The OOD detection results of DFB and its competing methods with CIFAR10 and CIFAR100 as in-distribution data are reported in Tabs. 1 and 2, respectively. Overall, DFB substantially improves four different SotA detection methods in all three evaluation metrics on both datasets, and obtains new SotA performance. We discuss the results in detail as follows.

**Enhancing Different OOD Detection Methods.** Four different SotA methods – MSP, ODIN, Energy, and ViM – are used as foreground-based OOD detection baseline models and plugged into DFB to perform joint foreground and background OOD detection. Their results are shown at the bottom of Tabs. 1 and 2.

Compared to all the four plug-in base models, DFB can significantly improve the performance of all evaluation metrics in terms

**Table 1: OOD detection results with CIFAR10 as in-distribution data. All methods are based on ID training data without using any external outlier data. [†] indicates that the results are taken from the original paper, and other methods are reproduced using the same network architecture. Four post-hoc foreground OOD detection methods are respectively plugged into our method 'X'-DFB, with improved results highlighted in red and in blue otherwise. The best result per dataset is boldfaced.**

| Methods | OOD Datasets | | | | Average |
| | CIFAR100 | SVHN | Places365 | Textures | |
| | FPR95↓ /AUROC↑ /AUPR↑ | | | | |
|---|---|---|---|---|---|
| MaxLogit [15] [ICML'22] | 39.11/85.07/78.13 | 17.95/94.78/84.22 | 24.05/91.10/86.41 | 7.93/97.57/98.07 | 22.26/92.13/86.71 |
| KL-Matching [15] [ICML'22] | 33.63/90.20/88.18 | 25.70/95.21/88.37 | 25.25/92.88/90.78 | 12.61/97.28/98.21 | 24.30/93.89/91.38 |
| ReAct [39] [NIPS'21] | 34.75/84.10/79.89 | 20.03/90.58/76.30 | 23.45/91.88/89.43 | 10.27/96.53/97.69 | 22.12/90.77/85.83 |
| MaSF[†] [13] [ICLR'22] | - /82.10/ - | - /99.80/ - | - /96.00/ - | - /98.50/ - | - /94.10/ - |
| DML+[†] [51] [CVPR'23] | 42.55/91.36/ - | 3.37/99.38/ - | 24.34/94.87/ - | 15.31/97.05/ - | 21.39/95.67/- |
| MSP [16] [ICLR'17] | 33.44/89.01/84.10 | 17.40/95.72/88.68 | 22.47/92.93/89.79 | 8.55/97.66/98.38 | 20.46/93.83/90.24 |
| MSP-DFB [Ours] | 23.75/94.29/93.48 | 2.55/98.94/97.88 | 5.05/98.49/98.60 | 0.02/99.90/99.95 | 7.84/97.90/97.48 |
| ODIN [27] [ICLR'18] | 34.62/87.83/81.92 | 16.13/95.66/87.30 | 22.15/92.43/88.59 | 7.45/97.86/98.37 | 20.09/93.45/89.04 |
| ODIN-DFB [Ours] | 22.15/95.50/95.29 | 4.27/99.19/98.30 | 8.08/98.66/98.76 | 0.34/99.92/99.95 | 8.71/98.32/98.07 |
| Energy [28] [NIPS'20] | 41.98/84.25/77.47 | 19.73/94.46/83.67 | 25.42/90.74/86.06 | 8.72/97.45/97.99 | 23.96/91.73/86.30 |
| Energy-DFB [Ours] | 19.90/94.98/93.76 | 3.10/99.28/98.14 | 6.96/98.60/98.52 | 0.53/99.87/99.91 | 7.62/98.19/97.58 |
| ViM [42] [CVPR'22] | 15.25/96.92/**96.78** | 1.27/99.47/99.08 | 2.74/99.32/99.34 | 0.11/99.93/99.96 | 4.84/98.91/98.79 |
| ViM-DFB [Ours] | **13.49**/**97.08**/96.75 | **0.41**/**99.85**/**99.68** | **0.72**/**99.85**/**99.85** | **0.00**/**100.00**/**100.00** | **3.65**/**99.20**/**99.07** |

**Table 2: OOD detection results with CIFAR100 as in-distribution data. The notations here are the same as that in Tab. 1.**

| Methods | OOD Datasets | | | | Average |
| | CIFAR10 | SVHN | Places365 | Textures | |
| | FPR95↓ /AUROC↑ /AUPR↑ | | | | |
|---|---|---|---|---|---|
| MaxLogit [15] [ICML'22] | 61.61/81.09/79.25 | 37.12/91.29/80.77 | 71.89/73.12/67.64 | 37.61/90.63/93.73 | 52.06/84.03/80.35 |
| KL-Matching [15] [ICML'22] | 64.49/79.54/74.46 | 47.86/89.08/76.63 | 73.55/78.04/76.61 | 46.63/88.97/92.16 | 58.13/83.91/79.96 |
| ReAct [39] [NIPS'21] | 70.81/79.62/78.97 | 53.00/88.88/78.43 | 82.64/68.11/63.28 | 52.80/88.15/92.58 | 64.81/81.19/78.31 |
| MaSF[†] [13] [ICLR'22] | - /64.00/ - | - /96.90/ - | - /81.10/ - | - /92.00/ - | - /83.50/ - |
| DML+[†] [51] [CVPR'23] | 79.35/76.69/ - | 21.69/96.51/ - | 68.31/83.31/ - | 49.24/88.56/ - | 54.65/86.27/ - |
| MSP [16] [ICLR'17] | 64.25/81.52/80.87 | 49.50/88.92/79.07 | 72.10/76.18/71.52 | 46.24/89.33/93.44 | 58.02/83.99/81.23 |
| MSP-DFB [Ours] | 58.76/84.67/84.25 | 50.75/89.27/82.39 | 67.82/85.20/88.06 | 28.21/95.86/97.85 | 51.38/88.75/88.14 |
| ODIN [27] [ICLR'18] | 59.67/82.39/80.79 | 38.11/91.32/81.40 | 69.80/75.39/69.81 | 37.38/91.10/94.22 | 51.24/85.05/81.55 |
| ODIN-DFB [Ours] | 55.92/87.31/88.02 | 32.79/90.60/76.03 | 55.34/81.56/79.62 | 10.78/97.40/98.23 | 38.71/89.22/85.48 |
| Energy [28] [NIPS'20] | 64.34/80.48/78.89 | 36.76/91.38/80.98 | 74.75/72.14/67.10 | 39.17/90.37/93.61 | 53.75/83.59/80.15 |
| Energy-DFB [Ours] | **54.02**/**88.12**/**88.82** | 24.78/93.39/81.94 | 48.87/85.72/82.67 | 7.11/98.41/98.92 | 33.70/91.41/88.09 |
| ViM [42] [CVPR'22] | 59.13/85.72/85.87 | 10.23/97.90/95.74 | 49.38/87.23/86.19 | 2.45/99.47/99.69 | 30.30/92.58/91.87 |
| ViM-DFB [Ours] | 60.88/85.74/86.38 | **7.58**/**98.40**/**96.39** | **20.93**/**96.06**/**96.23** | **0.16**/**99.96**/**99.98** | **22.39**/**95.04**/**94.74** |

of the average results over the four OOD datasets on both of the CIFAR10 and CIFAR100 datasets. In particular, for the averaged improvement across the four base models, DFB boosts the FPR95 by 10.38%, the AUROC by 3.92% and the AUPR by 6.96% AUPR in Tab. 1; and similarly, it boosts the FPR95 by 11.78% , the AUROC by 4.8% and the AUPR by 5.41% in Tab. 2. Note that even for the base model ViM, the most recent SotA method, DFB can still considerably enhance its performance, especially on some datasets where ViM does not work well, such as the SVHN, Places365, and Textures datasets, resulting in over 6% reduction in FPR95 and 2.5% increase in both AUROC and AUPR on the CIFAR100 data. DFB shows slight performance drops on some metrics of CIFAR and

SVHN OOD datasets, which are mainly caused by suboptimal combinations of the foreground and background OOD scores with the default temperature setting. The explanation would be discussed in detail using Fig. 6 in Sec. 4.2.

**Comparison to SotA Methods.** DFB is also compared with five very recent SotA methods, including MaxLogit [15], KL-Matching [15], ReAct [39], MaSF [13] and DML+ [51] , with their results reported at the top of Tabs. 1 and 2. Among all our four DFB methods and the SotA methods, ViM-DFB is consistently the best performer except the CIFAR10 data in Tab. 2 where Energy-DFB is the best detector. This is mainly because the ViM is generally the best semantic-feature-based OOD scoring method, and DFB can perform better when the plug-in base model is stronger. Further, it

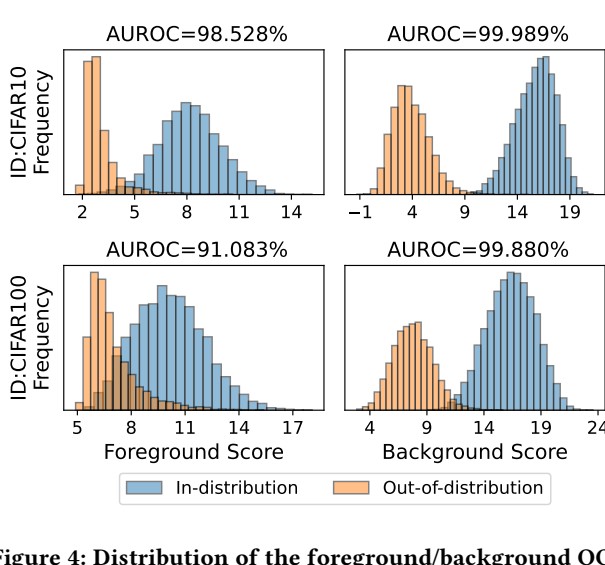

**Figure 4: Distribution of the foreground/background OOD scores of ID (CIFAR10/100) and OOD samples (Textures) in DFB.**

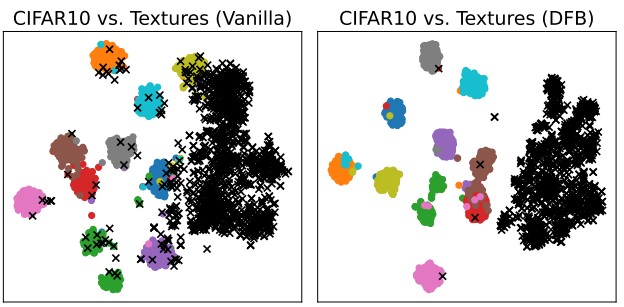

**Figure 5: t-SNE visualization of the features learned by the vanilla classification network and DFB, where the colored dots are ID samples of different classes, and the black × are OOD samples.**

is impressive that although the base models MSP, ODIN and Energy that largely underperform the SotA competing methods, DFB can significantly boost their performance and outperform these SotA competing methods on nearly all cases in Tabs. 1 and 2.

**The Reasons behind the Effectiveness of DFB.** We aim to understand the effectiveness of DFB from two perspectives, including the foreground and background OOD scoring, and the latent features learned in DFB, with the results on the Textures dataset reported in Figs. 4 and 5 respectively. We can see in Fig. 4 that the background OOD scores in DFB enable a significantly better ID and OOD separation than the foreground OOD scores, indicating that the ID and OOD samples can be easier to be separated by looking from the background features than the semantic features since there can be more background differences than the foreground ones in each ID/OOD image. From the feature representation perspective, compared to the features learned in the vanilla $K$-class classifier in

**Table 3: FPR95 Results of DFB and its variants.**

| Module | | BG | Energy | Energy-DFB | ViM | ViM-DFB |
|---|---|---|---|---|---|---|
| $S_h$ | | | ✓ | ✓ | ✓ | ✓ |
| $S_b$ | | ✓ | | ✓ | | ✓ |
| **CIFAR10** | **CIFAR100** | 38.16 | 41.98 | 19.90 | 15.25 | **13.49** |
| | **SVHN** | 2.60 | 19.73 | 3.10 | 1.27 | **0.41** |
| | **Places365** | 4.40 | 25.42 | 6.96 | 2.74 | **0.72** |
| | **Textures** | 0.04 | 8.72 | 0.53 | 0.11 | **0.00** |
| | **Average** | 11.30 | 23.96 | 7.62 | 4.84 | **3.65** |
| **CIFAR100** | **CIFAR10** | 89.24 | 64.34 | **54.02** | 59.13 | 60.88 |
| | **SVHN** | 26.61 | 36.76 | 24.7 | 10.23 | **7.58** |
| | **Places365** | 26.55 | 74.75 | 48.87 | 49.38 | **20.93** |
| | **Textures** | 0.53 | 39.17 | 7.11 | 2.45 | **0.16** |
| | **Average** | 35.73 | 53.75 | 33.70 | 30.30 | **22.39** |

Fig. 5 (left), the features learned by the $(K + 1)$-class classifier in DFB (Fig. 5 (right)) are more discriminative in distinguishing OOD samples from ID samples, which demonstrates that the classifier can learn better ID representation after disentangling foreground and background features.

## 4.2 Ablation Study

**Background OOD Score $S_b$ and the Joint OOD Score $S$.** Tab. 3 shows the FPR95 results of OOD scoring methods in our model, including the use of background OOD scores $S_b$ only (BG), foreground OOD scores $S_h$ (Energy and ViM are used), and the full DFB model (See Appendix C.1 for more detailed results). Compared to the two semantic OOD scoring methods, Energy and ViM, using only the background OOD scoring $S_b$ in DFB can obtain significantly reduced FPR95 errors, especially on OOD benchmarks such as Places356 and Textures where significant background differences are presented compared to the in-distribution background. This demonstrates that DFB can effectively learn the in-distribution background features that can be used to detect OOD samples from the background aspect. Nevertheless, BG works less effectively on the benchmark CIFAR100 vs. CIFAR10 where the background difference is weak and detecting OOD samples rely more on the foreground features. In such cases, the full DFB models – Energy-DFB and ViM-DFB – that synthesize semantic OOD scores $S_h$ and background OOD scores $S_b$ are needed; they significantly outperform the separate foreground/background OOD scoring methods across the datasets.

**Temperature $T$ in Synthesizing Foreground and Background OOD Scores.** One key challenge in plugging existing foreground OOD scores into DFB in Eq. (7) is the diverse range of different foreground OOD scores yielded by the existing methods. Fig. 6 the variants of DFB of using different temperature $T$ values to study the effects (see Appendix C.2 for the results on the other datasets). We can observe that the performance of all methods in CIFAR100 vs. CIFAR10 gradually improves as the temperature increases. This is because the increase of $T$ narrows down the distribution of background scores, thus making the final OOD scores emphasizing more on the foreground OOD scores, which are more effective in the OOD datasets like CIFAR100 (ID) vs. CIFAR10 (OOD) where the background difference is very small. In contrast, the performance of all methods in CIFAR100 (ID) vs. Places365 (OOD) gradually decreases

**Table 4: Top-1 accuracy results of in-distribution classification. Vanilla is the primitive trained classification network $\phi$ in Sec. 3.2.**

| Method | CIFAR10 | CIFAR100 |
|--------|---------|----------|
| Vanilla | 97.25% | 85.94% |
| DFB | 97.13% | 86.17% |

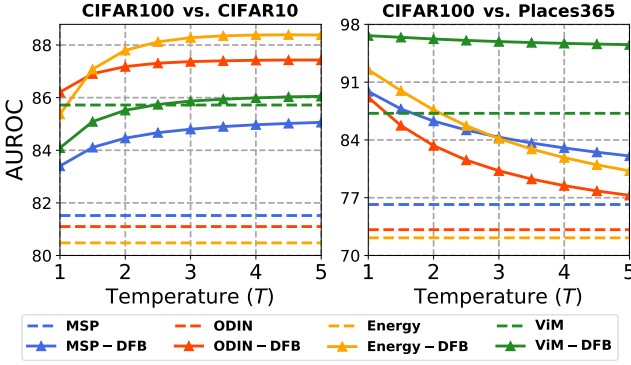

**Figure 6: AUROC results of DFB using varying $T$ settings.**

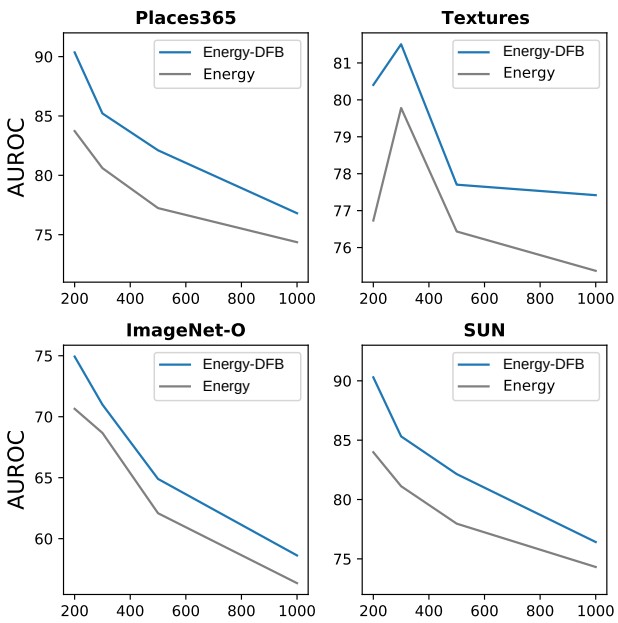

**Figure 7: AUROC results of DFB and the Energy baseline in large-scale semantic space using ImageNet-1k as ID data.**

as the temperature increases. This is because the background distribution difference dominates over the foreground difference in such cases, on which enlarging the distribution of background OOD scores is more effective. $T = 2.5$ is generally a good trade-off of the foreground and background OOD scores, and it is thus used by default in DFB. Note that adjusting $T$ generally does not bring the overall performance down below the baseline performance, showing the effectiveness of DFB using different $T$ values.

### 4.3 Further Analysis of DFB

**In-distribution Classification Accuracy.** A potential risk of modifying the primitive classification network for OOD detection is the large degradation of the in-distribution classification accuracy. As shown in Tab. 4, our proposed DFB does not have this issue, as DFB has only 0.12% top-1 accuracy drop on the CIFAR10 dataset and improves the classification performance by 0.23% on the CIFAR100 dataset. This result indicates that the dense prediction training in DFB ensures effective learning of foreground features, while learning the background features. The 0.23% accuracy increase on CIFAR100 also indicates that the dense prediction task can also improve the foreground feature learning for in-distribution classification.

**Extending to Large-scale Semantic Space.** A further challenge for OOD detection is on datasets with a large number of ID classes and high-resolution images, *e.g.*, ImageNet-1k [9]. Fig. 7 presents the detection performance of DFB using ImageNet-1k as in-distribution dataset and on four OOD datasets, including two new high resolution datasets, ImageNet-O [18] and SUN [46]. To examine the impact of the number of classes, we show the results using $C \in$

{200, 300, 500, 1000} randomly selected ID classes from ImageNet-1k ($C = 1000$ is the full ImageNet-1k data; see `Appendix B.2` for more details). The results show that DFB can consistently and significantly outperform its base model Energy with increasing number of ID classes on four diverse OOD datasets, indicating the effectiveness of DFB working in large-scale semantic space. On the other hand, as expected, both Energy and DFB are challlenged by the large semantic space, and thus, their performance decreases with more ID classes. Extending to large-scale semantic space is a general challenge for existing OOD detectors. We leave it for future work.

## 5 CONCLUSIONS

In this paper, we reveal the importance of disentangling foreground and background features in open-world classification and introduce background features for OOD detection that are neglected in current approaches. We further propose a novel OOD detection framework DFB that utilizes dense prediction networks to segment the foreground and background from in-distribution training data, and jointly learn foreground and background features. It then leverages these background features to define background OOD scores and seamlessly combines them with existing foreground-based OOD methods to detect OOD samples from both foreground and background aspects. Comprehensive results on popular OOD benchmarks with diverse background features show that DFB can significantly improve the detection performance of four different existing methods. Through this work, we promote the design of OOD detection algorithms to achieve more holistic OOD detection in real-world applications.

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
