# OpenReview forum: "Improving Open-World Classification with Disentangled Foreground and Background Features"
_acmmm.org/ACMMM/2024/Conference — MM2024 Poster_

### Official Review · Reviewer_ZoqK · 2024-05-24

**Rating:** 3
**Confidence:** 3

**Summary:**

This paper focuses on the problem of open-world classification and propose an new approach named Disentangle the Foreground and Background features (DFB) that is able to achieve out-of-distribution samples detection. The proposed DFB utilizes dense prediction networks to segment the foreground and background from in-distribution training data, and jointly learn foreground and background features. Then, the
background features is utilized to calculate the background OOD scores and then combined with the OOD scores from foreground features for OOD detection.The experimental results on some OOD benchmarks show the effectiveness of the proposed method.

**Strengths:**

1. The topic of OOD detection is important in community of computer vision and has some relations with the community of multi-media.
2. The idea of improving the accuracy of outlier detection by utilizing the background features generated by a module like CAM has some novelty.
3. The experimental results show the effectiveness of the proposed method in some extent.
4. The paper is generally written well and organized clearly. I do not find obvious typo or grammatical errors.

**Limitations:**

1. The concept of open-world in the title fails to exactly match the propose method. Indeed, this paper focuses on the problem of out-of-distribution detection or novelty detection. As explained in the following reference paper, the concept of open-world recognition usually includes not only outlier detection, but also incremental learning. However, I do not find any content about incremental learning for update the model in this paper. I suggest the authors change open-world into open-set.
Abhijit Bendale, Terrance Boult. Towards Open World Recognition. IEEE Conference on Computer Vision and Pattern Recognition, 2015.
2. The motivation of utilizing the background features to improve the performance of OOD detection has not been explained clearly. Please explain why the (K+1)-th dimension of the prediction logit vector yielded by $f_{\Theta_c}$ can be directly used as the OOD score. According to the training procedure, this score should reflect the similarity between the test sample and the background of the in-distribution samples. I do not quite understand the relationship between it and the OOD samples.
3. Considering that the capability of semantic segmentation for each in-distribution class is learned in a weakly-supervised manner, why DFB can solve the problem of spurious correlations with other objects caused by dataset bias? If the horse always appears with a person on it the training data, I do not expect the model can distinguish the horse and the person in the image without more supervised information. Please explain it.
4. In the third line of the first paragraph of section 3.1, "this type of approach" should be "approaches of this type".

**Suitability:**

2

---

### Official Review · Reviewer_dBEN · 2024-05-25

**Rating:** 4
**Confidence:** 3

**Summary:**

This paper ocuses on improving open-world classification by introducing a new framework that disentangles foreground and background features in ID training samples. This work leverages both foreground and background features to enhance the detection of OOD inputs in the open-world classification setting. The method is evaluated on CIFAR10 and CIFAR100 in-distribution datasets.

**Strengths:**

1. The paper proposes a method that enhances OOD detection by separating foreground features and background features, and provides a analysis of the motivation behind this approach.
2. The experimental results show the effectiveness of the method.

**Limitations:**

1. The paper constructs an OOD detection method from the perspective of foreground and background features, which is somewhat similar to work on visual interpretability. Can the authors analyze this from the perspectives of saliency analysis and visual attribution methods, and compare with some similar OOD detection methods using the similar methods?
2. The paper only evaluate the method on CIFAR-10 and CIFAR-100 datasets, and do not compare it with more recent works based on features. Additionally, can the method discussed in the paper also perform well in scenarios with higher resolution images, such as ImageNet-1K?

**Suitability:**

2

---

### Official Review · Reviewer_iACJ · 2024-05-28

**Rating:** 4
**Confidence:** 3

**Summary:**

This work proposes a new framework that separately extracts foreground and background features for open-world classification.

**Strengths:**

- The idea of the paper is easy to follow. The presentation is good.
- The improved performances are achieved compared to baselines.

**Limitations:**

- Is this work doing OOD detection or open-world recognition? As I learn, open-world learning involves incrementally learning or discovering new classes of data.
- Are “horse” and “person” unknown classes in the first column of Figure 1? There are not many differences between Vanilla and DFB-FG.
- The details of “Segment and Learn” in Figure 2 are missing.
- The results of more challenging datasets should be provided. The results of baselines are saturated.

**Suitability:**

2

---

### Meta-Review · Area_Chair_g7dp · 2024-07-04

**Recommendation:** Accept (Poster)
**Confidence:** 4

**Metareview:**

The paper investigates Out-of-Distribution (OOD) detection by separating foreground and background features, achieving promising performance. However, a primary concern is that the study mainly focuses on small-scale datasets in the main paper. The authors do provide supplementary results on the ImageNet dataset in the rebuttal. With two reviewers recommending acceptance and one suggesting rejection but not opposing acceptance, the AC recommends accepting the paper. The authors are advised to address the reviewers' comments in their rebuttal.